# Transcriptional Profiles of Murine Bone Marrow-Derived Dendritic Cells in Response to Peste des Petits Ruminants Virus

**DOI:** 10.3390/vetsci6040095

**Published:** 2019-11-29

**Authors:** Lingxia Li, Jinyan Wu, Dan Liu, Guoyu Du, Yongsheng Liu, Youjun Shang, Xiangtao Liu

**Affiliations:** State Key Laboratory of Veterinary Etiological Biology, Key Laboratory of Animal Virology of the Ministry of Agriculture, Lanzhou Veterinary Research Institute, Chinese Academy of Agricultural Sciences, Lanzhou 730046, China; lilingxia963@foxmail.com (L.L.); wujinyan@caas.cn (J.W.); liudan03@163.com (D.L.); duguoyu@163.com (G.D.); liuyongsheng@caas.cn (Y.L.); liuxiangtao@caas.cn (X.L.)

**Keywords:** peste des petits ruminants virus, bone marrow-derived dendritic cells, transcriptome, RNA sequencing

## Abstract

**Background:** Peste des petits ruminants virus (PPRV) is the causative agent of PPR, which can cause an acute, highly contagious and fatal disease of sheep and goats, resulting in significant economic losses for commercial animal husbandry due to its high mortality and morbidity. As professional antigen-presenting cells, dendritic cells (DCs) play a unique role in innate immunity. This study aimed to gain a deeper understanding of the transcriptional response of bone marrow-derived dendritic cells (BMDCs) stimulated with PPRV. **Results:** Transcriptional profiling was performed using RNA sequencing. Herein, we reported that compared to untreatedBMDCs, 4492 differentially expressed genes (DEGs) were identified following PPRV stimulation, out of these DEGs 2311 were upregulated and 2181 were downregulated, respectively. A total of three gene ontology (GO) term clusters of biological process, cell component and molecular function were significantly enriched in 963 GO terms in the PPRV-stimulated BMDCs. These GO clusters were related to inflammatory response, cell division and vacuole, anchoring junction, positive regulation of cellular component and nucleoside binding. Kyoto Encyclopedia of Genes and Genomes (KEGG) pathways of DEGs were enriched in a chemokine signaling pathway, protein processing in endoplasmic reticulum, cell cycle and mTOR signaling pathway. Additionally, identified DEGs of BMDCs were further validated by qRT-PCR and the results were in accordance with the change of the genes. This study suggested the effects of PPRV stimulation on the maturation and function of BMDCs. **Conclusion:** We found that the dramatic BMDCs transcriptome changes triggered were predominantly related to an inflammatory response and chemokine signaling pathway.

## 1. Background

Peste des petits ruminants virus (PPRV) is the causative agent of an acute, highly contagious disease that primarily infect small ruminants, especially goats and sheep [1,2,3]. The PPR disease causes a severe impact on the livelihood of low-income livestock keepers. It was reported that pigs can also be infected by PPRV [4], but most infections remain undetected. Moreover, some wild ungulates including Tibetan gazelle, African grey duiker and white-tailed deer are also susceptible to PPRV. In addition, some evidence suggest that PPRV is extending its host range, and an increasing number of wild and domestic animal species have been reported to be susceptible to PPRV over the past few decades. Therefore, proper control measures have become necessary to prevent their rapid spread throughout the world. PPRV belongs to morbillivirus (MV) of the paramyxoviridae family, which is an enveloped, non-segmented, negative-strand ribonucleic acid virus [5,6]. The immune response to morbillivirus is regulated by the innate and adaptive immune systems. In an adaptive immune response, pathogen invading organisms will activate helper T cells and secrete cytokines, which will stimulate the proliferation and differentiation of T cells. Additionally, it also activates other cells, including B cells, macrophages, and other lymphocytes. Immunological studies have largely focused on adaptive immune responses to PPRV infection and vaccination [7,8]. Like in other morbillivirus infections, PBMCs also play a major role in immune responses against PPRV infection [9]. Due to the effects of PPRV stimulation towards some other immune system remaining vague, including lymphocytes, dendritic cells, monocytes or macrophages, granulocytes and mast cells, we conducted this investigation about PPR virus against murine dendritic cells considering the problem of animal ethics and source of goat antibodies.

Dendritic cells (DCs), the most abundant immune cells, were derived from the blood and mainly differentiated from multi-functional stem cells. As the principal regulators of the immune system, DCs were mainly applied to antigen processing and presenting [10]. DCs were induced from bone marrow mononuclear cells and peripheral mononuclear cells. Major histocompatibility complex class II (MHC-II), costimulatory molecules such as CD86, CD80, CD83 and CD40, and chemokine receptors were highly expressed in mature dendritic cells (mDCs) [11]. mDCs can secrete interleukin (IL)-12 and their main function is to process and present antigens to T cells. Thereby stimulating T cells could produce large quantities of interferons (IFN). Furthermore, DCs were specialised in antigen-presenting cells (APCs) and played a pivotal role in the initiation of immune responses [12,13]. Importantly, a number of viruses infect DCs, modulating the immune response after infection with or without virus replication and has been performed until now [14,15,16].

Transcriptional sequencing technology has facilitated the development of veterinary molecular biology thus it has also become the frontier of biological and medical research [17,18,19]. Next-generation sequencing could illuminate the overall transcriptional activities of a given species at the single nucleotide level [20]. In the present study, we found that PPRV induced BMDCs responses including surface markers, cytokines and the uptake of FITC-dextran [21,22]. As DCs are professional antigen-presenting cells, in order to identify key regulatory factors during PPRV interaction with BMDCs at the transcriptional level, we investigated deep RNA sequencing of BMDCs stimulated with PPRV, which would be meaningful for identifying key genes or gene sets involved in a transcriptional response towards dendritic cells to PPRV stimulation and would provide insight into the immunological research.

## 2. Methods

### 2.1. Animals and Cells

Specific-pathogen-free (SPF) 8-week-old healthy female Balb/c mice were purchased from the Laboratory Animal Center of Lanzhou Veterinary Research (Lanzhou, China). In each independent experiment, six mice were selected from every group randomly (Mock, LPS, PPRV), a total of 18 mice for one independent experiment were used to get the tibia. Mice were fasted from 12:00 p.m. on the day before tibia collecting and the next morning, water was withheld. Firstly, we placed a cotton ball containing ether (2–4%) in a conical flask and put the mice in the flask and seal for 2 min. The mice were excited and then subdued, collapsed on its own. So mice were anaesthetized and they had no consciousness by inhaling ether, then mice could be put to a painless death and be used to isolate BMDCs. We conducted three independent repeats.

When the mice were anaesthetized and killed, 75% alcohol was used as a disinfectant for 5 min. The femur and tibia were removed, then cut off both ends of the bone and the internal bone marrow was washed with a 1 mL syringe by RPMI-1640 medium (Gibco, Carlsbad, CA, USA). The isolated cells were centrifuged for 10 min at 1000 rpm/min after filtration through a 300-mesh filter copper mesh. Then cells were lysed by erythrocyte lysate, washed three times with PBS containing 2% FBS and the cells were then cultured in RPMI-1640 medium containing 10% fetal bovine serum (FBS; Gibco, Grand Island, NY, USA), 100 units/mL penicillin, 100 μg/mL streptomycin, 20 ng/mL rmGM-CSF and 20 ng/mL rmIL-4 (Peprotech, Rocky Hill, NJ, USA) at a density of 1 × 10^6^ cells/mL, and incubated at 37 °C under 5% CO_2_. Half the medium was replaced every 2 days [23,24]. The experiments were performed in strict accordance with the Animal Ethics Procedures and Guidelines of the People’s Republic of China. All protocols were reviewed and approved by the Animal Research Ethics Committee of Lanzhou Veterinary Research Institute, Chinese Academy of Agricultural Sciences.

### 2.2. PPRV Inoculation

PPRV strain Nigeria 75/1, a vaccine virus was propagated by Vero cells in our laboratory. Vero cells were cultured in Minimum Essential Medium (MEM, Gibco, Carlsbad, CA, USA) medium containing 10% fetal calf serum (FCS) at 37 °C with 5% CO_2_. Then the Vero cells were seeded into T25cm^2^ flasks and inoculated with PPRV for 4–5 d. When cells showed 70–80% cytopathic effect (CPE), the virus was harvested by three cycles of freezing and thawing and stored at −80 °C. BMDCs were harvested on the seventh day and seeded in six-well plates (Corning, Corning, NY, USA). The experimental group were incubated with 300 μL PPRV/1 × 10^6^ cells for 24 h at 37 °C. RPMI-1640 medium treated cells were the mock control, LPS stimulated cells were the positive control.

### 2.3. Indirect Immunofluorescence Assay

The binding ability of PPRV that interacted with BMDCs was further confirmed by indirect immunofluorescence assay using PPRV polyclonal antiserum (PPRV-N polyclony antibody). BMDCs were seeded in a 35 mm dish, inoculated with PPRV for 24 h at 37 °C. After blocking overnight in 5% BSA the cells were incubated with diluted polyclonal antiserum (1:100) for 1.5 h at 37 °C. Then cells were washed with PBS three times and incubated with secondary antibody of anti-rabbit IgG (1:100, Proteintech, Wuhan, Hubei, China) conjugated with FITC for 1 h at 37 °C in dark place. Finally, green fluorescence was observed under a fluorescence microscope.

### 2.4. Library Construction and RNA Sequencing

The harvested cells were centrifuged at 1500 rpm for 10 min and stored at −80 °C for RNA extraction. RNA concentration was measured using a Qubit RNA Assay Kit and a Qubit 2.0 Fluorimeter (Life Technologies, Karlsbad, CA, USA). A total amount of 3 µg RNA per sample was used as input material for the RNA sample preparations. Then RNA integrity was assessed using the RNA Nano 6000 Assay Kit of the Bioanalyzer 2100 system (Agilent Technologies, CA, USA). Sequencing libraries were generated using an NEBNext Ultra RNA Library Prep Kit for Illumina according to manufacturer’s recommendations. Clustering of index-coded samples were performed on a cBot Cluster Generation System using TruSeq PE Cluster Kit v3-cBot-HS according to the manufacturer’s instructions [25,26]. After cluster generation, the library preparations were sequenced on an Illumina platform, and 125 or 150 bp paired-end reads were generated.

### 2.5. Data Processing and Transcriptome Analysis

Experimental samples were analyzed using RNA sequencing (RNA-Seq, Novogene Co, Ltd., Beijing, China). Then expressed genes in cells were annotated by bioinformatics analysis. By comparing our sequences with those in databases, mRNAs could be annotated into different categories [27]. To compare differentially expressed mRNAs between PPRV-stimulated and mock BMDCs. The two groups (three biological replicates for each group) were performed using the DESeq2 R package [28]. This software provided algorithms for determining differentially expressed genes (DEGs) based on the negative binomial distribution. The resulting *p*-values were adjusted using the Benjamini and Hochberg approach for controlling the false discovery rate (FDR). Genes with an adjusted *p*-value <0.05 and an absolute fold change of two were set as the threshold for significantly differentially expressed. Metabolic pathways of DEGs were analyzed using gene ontology (GO) and Kyoto Encyclopedia of Genes and Genomes (KEGG) [22]. We also used the clusterProfiler R package to test the statistical enrichment of DEGs in KEGG pathways of Novegene [29].

### 2.6. Verification of DEGs by qRT-PCR

The selected DEGs were further validated with the qRT-PCR method. Total RNA from PPRV-stimulated and untreated BMDCs were extracted by TRIzol Regent (Invitrogen, Carlsbad, CA, USA) using the standard protocol. The RNA concentration of each sample was measured with a NanoDrop ND-2000 spectrophotometer (Thermo Fisher Scientific, Waltham, MA, USA). First-strand cDNAs were synthesised using a PrimeScript RT reagent Kit with gDNA Eraser (TsingKe, Beijing, China). qRT-PCR was performed using a SYBR Green Primix Ex Taq Kit (TsingKe, Beijing, China) on a Mx3005p system (Agilent Technologies, Santa Clara, CA, USA) following the manufacturer’s instructions. Mouse β-actin was used as an internal reference gene. All primers were synthesised by Shenggong Co, Ltd., Shanghai, China. Each of the samples were repeated at least three times. The relative expression of mRNA levels were calculated using the 2^−∆∆CT^ method.

## 3. Results

### 3.1. Responses of BMDCs in Response to PPRV

According to our previous study [21], we found that 24 h was suitable for BMDCs differentiation. Compared with Mock, Lipopolysaccharide (LPS) (100 ng/mL) could activate DCs differentiation and increase the expression of CD80, CD40, CD86 and MHC-II while PPRV repressed this process (Figure 1A). The result of the indirect immunofluorescence assay showed that the PPRV-N protein can be detected through green fluorescence in the PPRV-stimulated group vs. Mock (Figure 1B), which indicated that BMDCs may have be interacted with PPRV by endocytosis and PPRV indeed existed in cells. To further identify key genes between PPRV-stimulated and untreated BMDCs, we conducted RNA sequencing technology. A large number of DEGs were identified by RNA-sequencing analysis. Compared with untreated BMDCs, 2311 genes were upregulated and 2181 genes were downregulated respectively following PPRV stimulation, which can be showed in the volcano plot (Figure 2).

### 3.2. Function Classification of the DEGs in Response to PPRV

A total of three gene ontology (GO) term clusters of biological process, cell component and molecular function were significantly enriched in 963 GO terms in the PPRV-stimulated BMDCs. DEGs were annotated to several GO categories belonging to the three branches of ontology−biological process (87.06%), molecular function (4.11%), and cell component (8.06%). These DEGs were involved in inflammatory response, cell division and vacuole, anchoring junction, positive regulation of cellular component, nucleoside binding and other signalling pathways. Each of the most significant 10 terms were listed in Figure 3. These findings indicated that the DEGs played important regulatary roles in the interaction of cell maturation or differentiation.

### 3.3. KEGG Enrichment of Key Genes Related to Signal Pathways

We further performed the functional classification of regulators to identify potential genes that underlay PPRV-induced immunomodulation. According to the KEGG enrichment, the validated targets of differentially expressed mRNAs were significantly enriched in 20 pathways (Figure 3D and Figure 4), such as protein processing in the endoplasmic reticulum, apoptosis, the HIF-1 signalling pathway and chemokine signalling pathway. We thus demonstrated that the identified DEGs may play important regulatory roles in cell structures, cell proliferation and differentiation.

### 3.4. Validation of DEGs by Quantitative RT-PCR

To further validate the DEGs of RNA sequencing analysis, several DEGs playing important roles in immune regulation were validated by quantitative real time polymerase chain reaction (qRT-PCR). mRNA-specific forward and reverse primers of each genes were listed in Table 1. qRT-PCR results showed that mRNA levels of Pfn2, Pf4, Gm5438, Cd14, Fbp1, Atp5e and Romo1 were increased by 6.33, 5.19, 8.8, 6.09, 7.49, 5.42 and 5.88-fold, respectively (Figure 5A). Then mRNA levels of Tarsl2, Bcap31, Il1b, Cstb, Klk1b11, Rapgef3, Cd9 were increased by 3.88, 2.19, 2.49, 3.34, 2.75, 3.24 and 1.48-fold, respectively, at 24 hpi. In addition, Cmklr1, Hal decreased by 0.13 and 0.23-fold. Others decreased about 0.4–0.6-fold (Figure 5B), which was in accordance with the RNA sequencing results only Fblim and Sam yielded inconsistent results between the two methods, although few variations were observed for these genes and above 90% of the results were reliable. We consistently found markedly more upregulated mRNAs than downregulated mRNAs.

## 4. Discussion

Previous studies have suggested that PPRV replication was tightly related by the innate immune system [30,31]. In this study, we used deep sequencing to identify cellular mRNAs, which were significantly altered in BMDCs in response to PPRV. Furthermore, according to our previous study, PPRV interaction with BMDCs could inhibit the phenotypic maturation of BMDCs by reducing the expression of CD86, CD40 and CD80 compared with mock-treated BMDCs. Meanwhile, a low titer of PPRV increased the secretion of cytokines including IL-6, IFN-γ and IL-10 [22], demonstrating that PPRV might induce immunoregulatory effects through modulating immune functions of BMDCs. With the advances in next-generation sequencing technology RNA-Seq has been widely used in veterinary science. In this study, we compared DEGs identified in BMDCs after PPRV stimulation. We found a number of DEGs (2311 upregulated and 2181 downregulated) along with PPRV stimulation. Similar research was conducted by Manjunath et al., which investigated PBMCs isolated from goat infected with PPRV. Their transcriptome data revealed 985 DEGs at 120 hpi, of which 117 exhibited a significant enrichment in immune system processes [32]. They also found that glyceraldehyde-3-phosphate dehydrogenase (GAPDH) was the most stable reference gene in PBMCs infected with PPRV [33]. As in our data, the selection of murine DCs might be a new exploration to investigate virus interaction with antigen-presenting cells.

Our results also showed that BMDCs responded by activating its immune response genes and its signalling pathways as the stimulation progressed. A previous study reported upregulation of Alox12 in colorectal tumors, which is involved in inflammation [34]. Inflammasome is essential for the organism in controlling disease. In inflammatory responses, IL-1β conducted adaptive immune responses by inducing the expression of immunity associated genes. Thus, it was possible that Alox12 may have also been involved in inflammatory responses in BMDCs after stimulation with PPRV. In addition to its function in innate immunity, proliferation can decrease in live, nondividing cells as in senescence. By contrast, Reep5 was downregulated at the transcriptional level. GO analysis indicated that Reep5 was related to cell differentiation, cell proliferation and the immune system, which suggested that differentiation and proliferation of BMDCs may be inhibited when stimulated with PPRV [35]. This might be caused by a reduction in Reep5, which will require further research in our later program.

Additionally, apoptosis pathways are related to cell death, which can also be thought of as one of the defense mechanisms for virus replication. Ctse and Cd9 were related to the innate immune system, but only Bcap31 regulated apoptosis. Bcap31 was investigated to determine its involvement in CASP8-mediated apoptosis [36,37]. PPRV infected with goat-PBMCs induced apoptosis, which implied that virus replication could be inhibited by killing cells [38]. In addition, apoptosis was related to interactions between viruses and the host immune system [39]. Hence, we observed the significantly upregulated Bcap31 transcript, indicating that PPRV interacted with BMDCs to induce apoptosis in cells to some extent. Additionally, Plcb1 played an important role in cell signal transduction, while most of the other proteins were glycoproteins related to actin synthesis, proteolysis and metabolism. At the transcriptional level, this gene was upregulated and accounted for the majority. This suggested that the abundance of genes involved in these pathways was less affected by PPRV. We selected a few genes that we were interested in to detect mRNA levels through qRT-PCR, which showed identical results to transcriptome sequencing analysis. Thus, our findings might help elucidate the cellular events that occurr in BMDCs following PPRV stimulation.

From these data it is obvious that the abundance of numerous genes that we have identified are regulated differently from the corresponding mRNAs. This has promoted us to examine more gene functions towards pathways. These functions and associated metabolic pathways provide an immunological basis for understanding the impact on the maturation and differentiation of BMDCs. In particular, other DEGs related to immunity regulation or viral infection need to be further investigated [40]. Based on bioinformatics analysis, mRNA levels represent an intermediate state between gene expression and protein expression. Our findings also indicated an important regulatory role of mRNAs in PPRV-stimulated BMDCs. The resulting data provides a roadmap for future studies that could lead to illustrating the cellular responses towards PPRV [41,42,43]. Our study provides a basic foundation for the case that BMDCs are used to study immune functions for many viruses.

## 5. Conclusions

In conclusion, our results have provided a comprehensive analysis of the responses of BMDCs stimulation with PPRV via transcriptome approaches, which revealed 4492 DEGs, among which 2311 genes were upregulated and 2181 genes were downregulated. We speculated that PPRV might repress the maturation and differentiation of PPRV-stimulated BMDCs through a series of immune-related genes. Bioinformatics analysis also showed that the majority of targeted DEGs were significantly enriched in immune processes, including genes related to the inflammatory response and protein processing in endoplasmic reticulum cell division as well as chemokine signalling pathway and so on. All in all, it showed that PPRV could elicit a response with BMDCs. We foresee that the data reported in this study will provide a valuable resource for the cellular processes in antigen processing and presenting.

## Figures and Tables

**Figure 1 vetsci-06-00095-f001:**
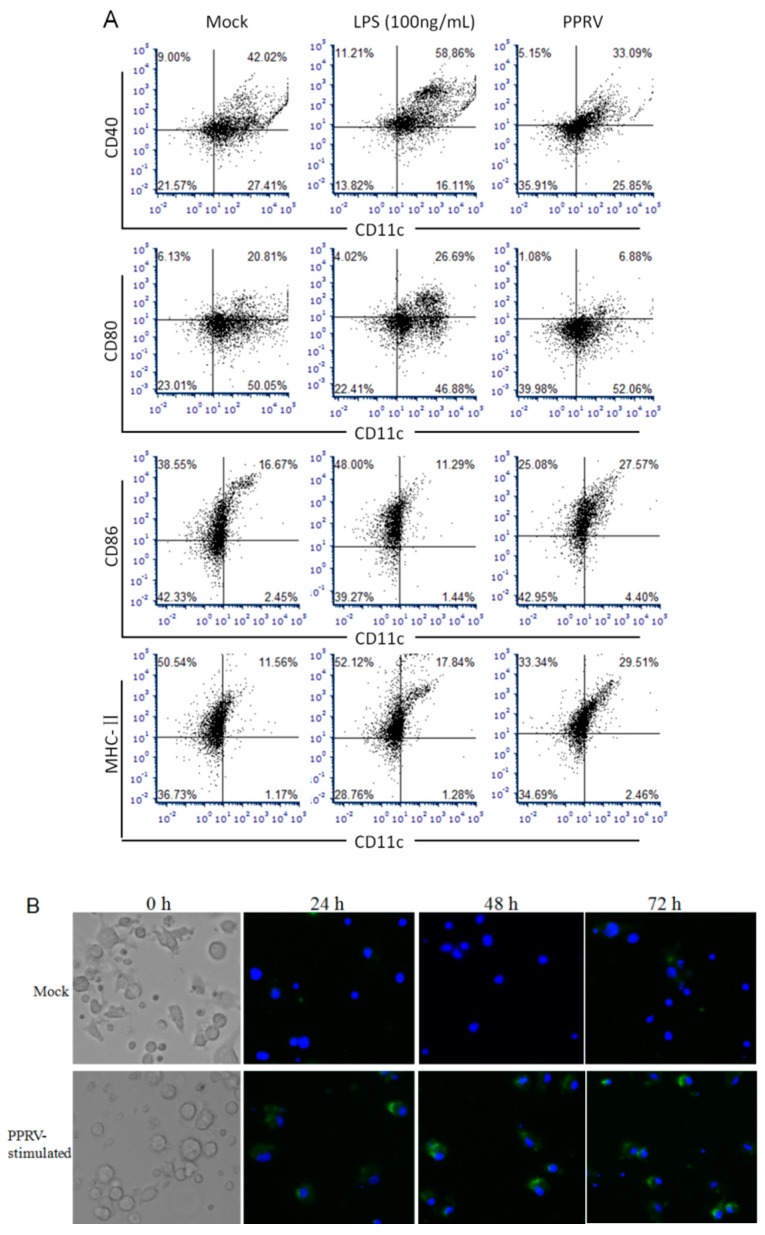
Confirmation of peste des petits ruminants virus (PPRV) interaction with bone marrow-derived dendritic cells (BMDCs). (**A**) BMDCs were treated with LPS (100 ng/mL) and PPRV (300 μL/L × 10^6^ cells), the levels of CD80, CD40, CD86 and MHC-II were detected by flow cytometry; (**B**) PPRV-N protein can be detected through green fluorescence in the PPRV-stimulated group vs. mock in indirect immunofluorescence assay.

**Figure 2 vetsci-06-00095-f002:**
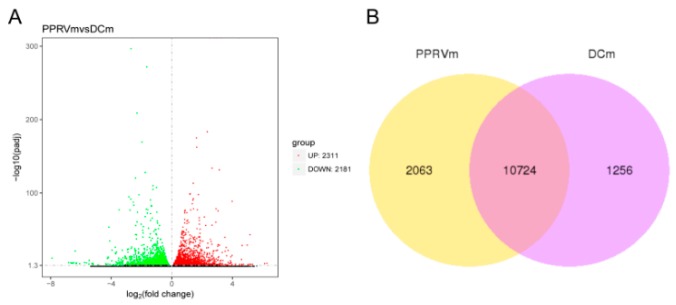
Differential expressed genes (DEGs) in response to PPRV. (**A**) Volcano map of DEGs. Red dots represent upregulated genes, and green dots show downregulated genes. The abscissa indicates variation in gene expression between different samples (log2 Fold Change), and the ordinate indicates the significance of the expression differences (−log10 padj). (**B**) Venn graph of DEGs.

**Figure 3 vetsci-06-00095-f003:**
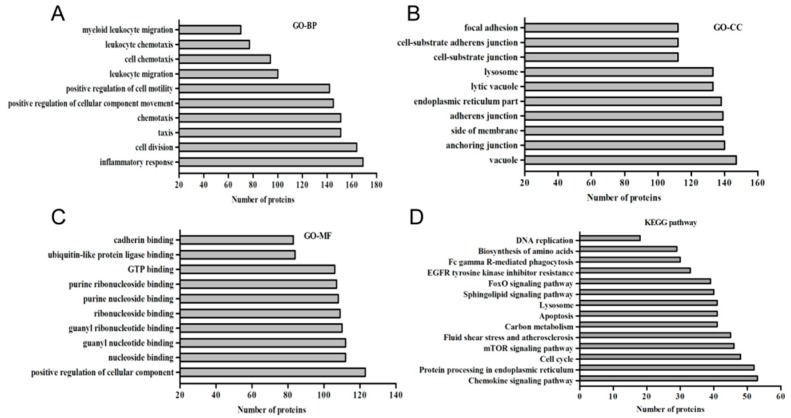
Function classification based on GO enrichment analysis of DEGs in response to PPRV. Graph of the proportions of GO enrichment results. The top 10 significant terms were shown on the graph. (**A**) Analysis of cellular component (GO-CC); (**B**) Analysis of cellular component (GO-BP); (**C**) Analysis of cellular component (GO-MF). (**D**) KEGG pathway enrichment analysis of DEGs in response to PPRV.

**Figure 4 vetsci-06-00095-f004:**
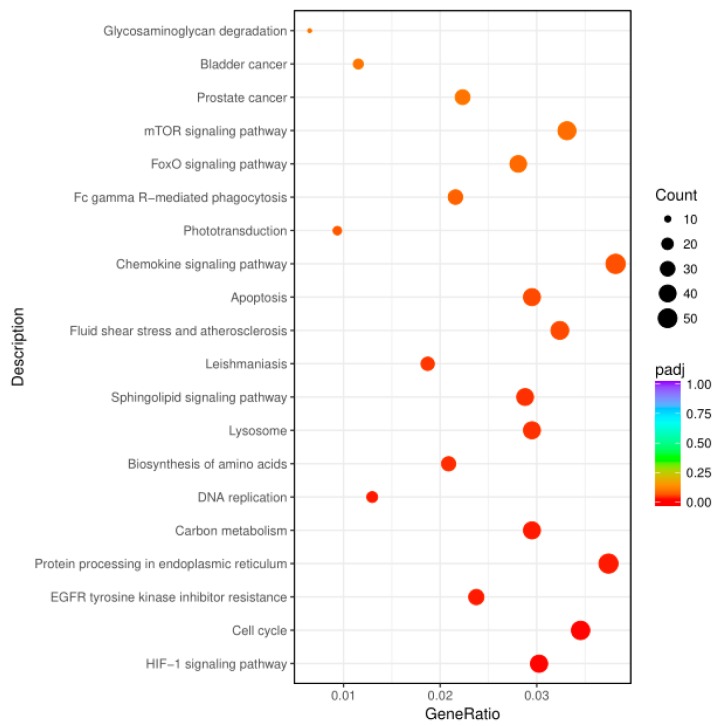
Bubble diagram enrichment analysis of DEGs in response to PPRV. The top 20 significant enriched pathways were shown on the graph.

**Figure 5 vetsci-06-00095-f005:**
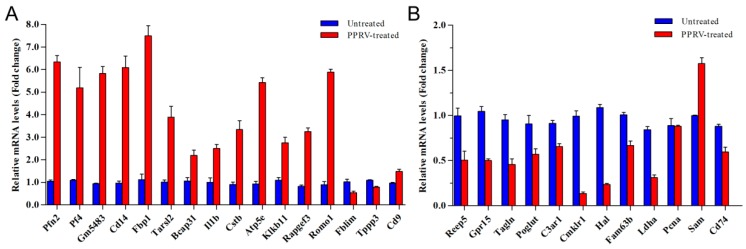
Validation of DEGs by qRT-PCR. (**A**) qRT-PCR analysis of upregulated genes; (**B**) qRT-PCR analysis of downregulated genes, the abscissa is gene name, the ordinate is CT values (n = 6).

**Table 1 vetsci-06-00095-t001:** Sequences of primers used for the analysis of gene expression by qRT-PCR.

Gene Name	Primer Sequence (5′ to 3′)	Product Size (bp)
Pfn2	F: GAAGGTTTCTTTACCAACGGTT	81
	R: GTCACCATCAACGTATAGGCTA	
Pf4	F: TCTCCTCTGGGATCCATCTTAA	114
	R: GGCAAATTTTCCTCCCATTCTT	
Gm5483	F: ATCCAGGAGATTGCTGATAAGG	81
	R: CTCAACAGCTTTGAACACTTCA	
Cd14	F: GACCTTAGTCACAATTCACTGC	87
	R: GAAAGACAGATTGAGCGAGTTT	
Fbp1	F: GATCTTTTTATACCCCGCCAAC	93
	R: GCCTTCTCCATGACATAAGCTA	
Tarsl2	F: CGATGTGGACTTAGATGACAGT	104
	R: TGATCTTTTCCTTTTCGCCAAC	
Bcap31	F: CTTTGTGGTTCTCATCGTCATC	92
	R: GGTTCACCTTTTCTGTCACATC	
Il1b	F: TGAATGAAGCACCAGCACAT	140
	R: AGCCTCATGGCCCAATTTCT	
Cstb	F: TCTGATTCGGGGCTCCTTTG	74
	R: TCCCTTCTCTCCATCACCGA	
Atp5e	F: CGACAGGCTGGACTCAGCTA	115
	R: TTATGCTGCTGCCCGAAGTC	
Klk1b11	F: AGTCTCGAATTGTTGGAGGATT	187
	R: GTTCACGTTGGAATAGCTTTGT	
Rapgef3	F: AACTTGCTGAGGGAACAGTATC	84
	R: CTTGGTCTGAGGAGATACGTTC	
Romo1	F: CTGTCTCAGGATCGGAATGC	103
	R: CATTCCAATGGCCATGAAAGTG	
Fblim1	F: CTCCATCAAAGGGATCGTCTG	108
	R: CTTCTCTCTCTGGAAGTGTGAC	
Tppp3	F: TGACGGACACCAGTAAGTATAC	125
	R: GTTTTTGTAGGCACTCACGTAG	
Cd9	F: ATCAAATACCTGCTCTTCGGAT	83
	R: AATCGGAGCCATAGTCCAATAG	
Gpr15	F: CACATCTGTCTTCCTCCCTATC	118
	R: GATGTCGATCAATCTTCGGTTG	
Reep5	F: CATCTCAATGAAAGCCATCGAG	88
	R: ATGCTGAACACACCATATACCA	
Tagln2	F: GTAAAGAAGATCCAGGCCTCTT	136
	R: TGTTCTTTCCTTCCCAGAGATC	
Poglut1	F: TAAAGCCATGGGTTCACTACAT	262
	R: AGTTCAGTTTTCAAACGTCTGG	
Pcna	F: GAAGTTTTCTGCAAGTGGAGAG	107
	R: CAGGCTCATTCATCTCTATGGT	
C3ar1	F: GTGCAAACTTATCCCATCCATC	92
	R: GTACTATCAGACATCGGTCCAG	
Hal	F: CAACGTCTTAGCCAAAGGTTAC	230
	R: GTCCATGGGCTTCTAGAACATA	
Cmklr1	F: GCGTCTTCCTGCTGACTGTCATC	85
	R: CGGATGCTGCGGTGGTTCTG	
Ldha	F:AAGACTACTGTGTAACTGCGAA	114
	R: ACTTGAAGATGTTCACGTTTCG	
Fam63b	F: CAAAGAAACTCCAGGAAGAGGA	198
	R: CTTTGTCTTTTTCCCGAGGTTC	
Cd74	F: GTGAACTGGAAGATCTTCGAGA	115
	R: ACTTGGTCAGTACTTTAGGTGG	
β-actin	F: AGTGTGACGTTGACATCCGT	126
	R: GCAGCTCAGTAACAGTCCGC	

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
