# Peer review of "Transcriptional Profiles of Murine Bone Marrow-Derived Dendritic Cells in Response to Peste des Petits Ruminants Virus"

_vetsci, 2019, doi:10.3390/vetsci6040095_

Round 1
Reviewer 1 Report
Li and colleagues (VETSCI-646855) report on a molecular biology study to investigate the transcriptional changes in bone marrow derived dendritic cells during Peste des petits ruminants virus (PPRV) infection. The investigators present evidence that their method detects alterations in transcripts after infection with PPRV of dendritic cells (MOI=1). Based on these findings, the authors claim that PPRV might repress the maturation and differentiation of PPRV-treated BMDCs through a series of immune-related genes (line 225-226).
In general, this is a properly developed and interesting study that provides an extensive transcription profile of PPRV infected dendritic cells. This paper will be of interest to scientists in PPRV research. Despite my general enthusiasm for the overall approaches and conclusions drawn, there were a few concerns and comments that came up during the review.
Main points:
The figure in 1B is published in Li et al. Veterinary Science from Elsevier (https://www.sciencedirect.com/science/article/pii/S0034528819300104). What is a vaccine virus? Why use this specifically? Please reword Animals and Cells section for more scholarly style of writing. First section appears to lack editing. why not use the term “infect” instead of stimulate when cells are positive with PPRV antibody? Also Why are there lesser cells in 24 h onwards vs 0 hrs? Please provide a more informative Figure legend describing the experiments in brief (e.g. what is seen in the FACS? What is the green signal in IHC?) as done in Figure 2. Also, what is rational for LPS? Before saying that the pathways related to DEG are probably important in a phenotype, there has to be an explanation of details provided. For instance, apoptosis pathways are related to cell death; which is the opposite of proliferation. Proliferation is not mutually exclusive with death since proliferation can decrease in live, nondividing cells as in senescence. This is why details are important to avoid confusion. Please indicate basis of significance in figure legend even if mentioned in methods. This is standard in scientific writing. What are GM5438, Kikb11, Rpgef3 genes?Why is Sam gene in down regulated due to PPRV when it is increased? Why are the genes CD40, 80, 86 not down regulated in current manuscript? “Bioinformatics analysis of RNA-seq can identify and annotate pathogen genes.”=no pathogen genes were mentioned in the text so why mention this in discussion? Where is Alox12 in the results to warrant the mention of this gene in discussion? If not in main figure but instead in total list of DEG, why mention in text of not in primary data in text? Solution is to provide complete DEG list with manuscript. Induction of apoptosis during virus infection is not always antiviral. Some viruses like influenza require activation of apoptosis for enhancement of viral replication. What does this sentence mean? “At the transcriptional level, this gene was up-regulated accounted for the majority.” “These functions and associated metabolic pathways provided an immunological basis for understanding how PPRV affect the maturation and differentiation of BMDCs.” Although authors may have shown the observation of differentiation after infection with PPRV, to conclude that these DEGs are the basis for the differentiation requires manipulation of these genes and evaluation of impact upon differentiation, which the current paper do not perform. Rephrase statement if this is not the claim. “This research will provide a basic foundation for exploring BMDCs as a cell model for investigating the immune functions [38-40].” Is this not the case where BMDCs are used to study immune functions for many viruses? The current work is novel in that it looks into PPRV, so must state this instead of a blanket statement. Correct for grammar: In conclusion, our results showed that PPRV could (elicit a) response with BMDCs. “PPRV might repress the maturation and differentiation of PPRV-treated BMDCs through a series of immune-related genes.” Again, correlation is not causation. No data showing causation as mentioned in previous comment. Are study participants referring to authors? Study participants are often study subjects so for this case it would be animals. Change to All Authors to avoid confusion that animals wrote consent.
Minor points:
What is DGE? Is DEP same as DEG? Why use different terms? Define early on in text to avoid confusion. What is DEPs? ontologe? What is “important regulation rules…” The word rule is ambiguous and connotes an agreed convention among people and not necessarily a principle. Rephrase ambiguous expression. Also correlation is not automatically causation (i.e. changes in transcript are causing phenotype). Need to perform direct manipulation to demonstrate cause. So is DEG the same as DEP? Please be consistent in terminology. Spell out quantitative RT-PCR? Can also put primer sequences in methods instead of a separate figure in text depending on Editor preference.
Author Response
Reviewer #1
Main points:
The figure in 1B is published in Li et al. Veterinary Science from Elsevier (https://www.sciencedirect.com/science/article/pii/S0034528819300104). What is a vaccine virus?
Answer for Q: We call it vaccine virus because this PPR virus was propagated from a vaccine strain Nigeria/75/1 while not wild virus. Besides, we changed Figure.1 as using another analysis model in flow cytometry. Figure.1B is another independent as we have published. In order to illustrate PPRV interacted with BMDCs we add this picture too.
Why use this specifically?
Answer for Q: Because the most widely used PPRV vaccine strains are Nigeria 75/1 (lineage II) and Sungri 96 (lineage IV), which provide complete protection across genetic lineages. In our lab, a stable passage strain Nigeria/75/1 was gained.
Please reword Animals and Cells section for more scholarly style of writing.
Answer for Q: Thanks a lot for reviewers' comments. Animals and Cells section was reworded. (See more details in lines 76-80, 85-90)
First section appears to lack editing. why not use the term “infect” instead of stimulate when cells are positive with PPRV antibody?
Answer for Q: Thank you for your useful suggestions. Because in our previous study, PPRV was an external irritant or antigen to BMDCs, but PPRV could cause some phenotypic changes in BMDCs so we did further research in this paper. The first section has been edited. Besides, although PPRV interacted efficiently with DCs, there was no evidence for virus replication, but the PPR virus did persist in BMDCs without loss of infectivity nor the induction of cell death (published paper https://doi.org/10.1016/j.rvsc.2019.06.011). So stimulate is more scholarly than infect.
Also Why are there lesser cells in 24 h onwards vs 0 hrs? Please provide a more informative Figure legend describing the experiments in brief (e.g. what is seen in the FACS? What is the green signal in IHC?) as done in Figure 2.
Answer for Q: Because BMDCs were isolated from murine bone, the cells were suspension cultured cells not adherent cells, then after fixed with 4% paraformaldehyde and washed with PBS each step in indirect immunofluorescence assay, the green fluorescence in suspended cells will be decreased in 24 h onwards vs 0 hrs. We have provided a more informative Figure legend describing the experiments in brief. (lines 164-167).
Also, what is rational for LPS?
Answer for Q: Thank you for your caution. LPS is applicable to the activation of B cells in mice, LPS (100ng/mL) could activate DCs differentiation and increase the expression of CD80, CD40, CD86 and MHC-Ⅱ compared with Mock control, it was a positive control in this study.
Before saying that the pathways related to DEG are probably important in a phenotype, there has to be an explanation of details provided. For instance, apoptosis pathways are related to cell death; which is the opposite of proliferation. Proliferation is not mutually exclusive with death since proliferation can decrease in live, nondividing cells as in senescence. This is why details are important to avoid confusion.
Answer for Q: Thanks for reviewers suggestion. The statement has been rephrased in this paper. (See more details in lines 235-240)
Please indicate basis of significance in figure legend even if mentioned in methods. This is standard in scientific writing. What are GM5438, Kikb11, Rpgef3 genes? Why is Sam gene in down regulated due to PPRV when it is increased? Why are the genes CD40, 80, 86 not down regulated in current manuscript?
Answer for Q: Thank you very much for suggestions. We have noticed this problem and have indicated basis of significance in figure legend. “GM5438”: also named Cstdc4, EG433016 and mCG_130182. Its protein name is “cystatin domain-containing 4”. “Kikb11”: it is Klk1b11,also named Klk1b11, Klk-11, Klk11. Its protein name is Kallikrein 1-related peptidase b11. “Rpgef”: it is rapgef3. Also named Rapgef3 Epac, Epac1. Its protein name is Rap guanine nucleotide exchange factor 3.
“Bioinformatics analysis of RNA-seq can identify and annotate pathogen genes.”=no pathogen genes were mentioned in the text so why mention this in discussion? Where is Alox12 in the results to warrant the mention of this gene in discussion? If not in main figure but instead in total list of DEG, why mention in text of not in primary data in text? Solution is to provide complete DEG list with manuscript. Induction of apoptosis during virus infection is not always antiviral. Some viruses like influenza require activation of apoptosis for enhancement of viral replication. What does this sentence mean?
Answer for Q: Thanks for reviewers' comments. Its our mistake that we have deleted wrong sentences. (See more details in lines 223-225)
“At the transcriptional level, this gene was up-regulated accounted for the majority.” “These functions and associated metabolic pathways provided an immunological basis for understanding how PPRV affect the maturation and differentiation of BMDCs.” Although authors may have shown the observation of differentiation after infection with PPRV, to conclude that these DEGs are the basis for the differentiation requires manipulation of these genes and evaluation of impact upon differentiation, which the current paper do not perform. Rephrase statement if this is not the claim.
Answer for Q: Thanks for reviewers' comments. We have rephrased statement in this paper. (See more details in lines 245-252)
“This research will provide a basic foundation for exploring BMDCs as a cell model for investigating the immune functions [38-40].” Is this not the case where BMDCs are used to study immune functions for many viruses? The current work is novel in that it looks into PPRV, so must state this instead of a blanket statement.
Answer for Q: Thanks for reviewers' comments. It is the case where BMDCs are used to study immune functions for many viruses. We have stated this instead of a blanket statement (line 259-261).
Correct for grammar: In conclusion, our results showed that PPRV could (elicit a) response with BMDCs. “PPRV might repress the maturation and differentiation of PPRV-treated BMDCs through a series of immune-related genes.” Again, correlation is not causation. No data showing causation as mentioned in previous comment. Are study participants referring to authors? Study participants are often study subjects so for this case it would be animals. Change to All Authors to avoid confusion that animals wrote consent.
Answer for Q: Thank you for reviewers' comments. We have changed participants to all authors (line 286) and the conclusion part has been reworded.
Minor points:
What is DGE? Is DEP same as DEG? Why use different terms? Define early on in text to avoid confusion. What is DEPs? ontologe? What is “important regulation rules…” The word rule is ambiguous and connotes an agreed convention among people and not necessarily a principle. Rephrase ambiguous expression. Also correlation is not automatically causation (i.e. changes in transcript are causing phenotype). Need to perform direct manipulation to demonstrate cause. So is DEG the same as DEP? Please be consistent in terminology.
Answer for Q: “DEG” it was abbreviation of differentially expressed genes in this study. It is not same as DEP. “DEP” it was differentially expressed proteins. We got it wrong, it is DEG not DGE or DEP and we correct it in manuscript. So it is consistent in terminology. “important regulation rules…” it means “important regulatary roles” It was our spelling mistake. We feel sorry and correct it.
Gene Ontology (GO) is a widely used Ontology in the field of bioinformatics, which covers three aspects of biology: cell constituents, molecular functions and biological processes.
Spell out quantitative RT-PCR? Can also put primer sequences in methods instead of a separate figure in text depending on Editor preference.
We have spelled out quantitative RT-PCR in the paper. There are 28 pairs of primers in this study, put all primer sequences in methods will take up a lot of pages in the paper. So we put them in a table.

Reviewer 2 Report
Dear Authors,
very good work, very good at you methods and the presentation of it. The results part is also very good. The introduction is very poor. Add please the why your research is very important for the reader.

Author Response
Reviewer #2
1, Add please more information in the background section is very short.
Answer for Q: Thank you very much for your useful suggestions. The background has been revised in the vetsci-646855R1. (See more details in lines 36-41, 44-52)
2, Methods are very good but be more specific 63-69, 81-87, 89-93, explain more the methods.
Answer for Q: Thank you very much for reviewers' comments. The methods have been rewrited more details in 2.1, 2.2 and 2.3 parts.

Reviewer 3 Report
This manuscript represents an important initial step in our understanding of the cellular response to PPRV infection and the resulting data provide a roadmap for future studies that could lead to prophylactic or therapeutic treatments for his important animal virus. I especially like the various formats used to present and illustrate the data, complex as it is. Also, the decision to test the validity of the DEGS by qRT-PCR and the resulting strong confirmatory data are considered strengths. After some editorial attention to English grammar and word usage, the manuscript is considered acceptable for publication.
In this manuscript, the authors use deep RNA sequencing to generate a profile of the transcriptional response in dendritic cells to PPRV infection. They identify almost 4,500 differentially expressed genes in these cells, 2,311 of which are upregulated and 2,181 are downregulated. These genes were involved in a variety of cellular processes, including inflammatory response, cell division, chemokine signaling, ER protein processing and cell cycle.
Strengths of the manuscript include the validation of many of the differentially expressed genes by qRT-PCR and the highly illustrative formats used in the figures to present the very complicated data. The only weakness is that the study does not result in any definitive newly identified gene that can be targeted in subsequent studies.
However, the manuscript does constitute an important initial step in our understanding of the cellular response to PPRV infection and the resulting data provide a roadmap for future studies that could lead to prophylactic or therapeutic treatments for this important animal virus. Requiring only some relatively minor editorial attention to English grammar and word usage, the manuscript is considered very strong.
Author Response
Reviewer #3
1, Strengths of the manuscript include the validation of many of the differentially expressed genes by qRT-PCR and the highly illustrative formats used in the figures to present the very complicated data. The only weakness is that the study does not result in any definitive newly identified gene that can be targeted in subsequent studies.
Answer for Q: Thanks for reviewers' comments. In our study, the selection of murine DCs maybe is one of our new innovation to investigate virus interaction with antigen-presenting cells.
2, However, the manuscript does constitute an important initial step in our understanding of the cellular response to PPRV infection and the resulting data provide a roadmap for future studies that could lead to prophylactic or therapeutic treatments for this important animal virus. Requiring only some relatively minor editorial attention to English grammar and word usage, the manuscript is considered very strong.
Answer for Q: Thanks a lot for reviewers' comments. The English grammar and word usage have been edited by a English native colleague.

Reviewer 4 Report
Li et al present an important study on a relvant veterinary pathogen. They investigate the transcriptome Profile of bone marrow derived immune cells in Response to peste des Petits ruminants Virus.
While the Overall study merits publications, some technical Details have to be clarified in advance. The authors use Balb/c mice. Although this is an inbred strain they should Show what the transcriptional priofile Alteration between several non-infected animals is in order to define thresholds. This is not or not sufficiently described.
Moreover it can make a difference if the mice are 6 or 8 weeks old, thus for transcriptome analyses all controllable Parameters should remain constant.
The reviewer also has not found Information on the number of animals in each Group.This is also essential for the study. It is strongly recommended that an a priori animal number claculation is to be performed to justify the Group size.
Author Response
Reviewer #4
1, Li et al present an important study on a relvant veterinary pathogen. They investigate the transcriptome Profile of bone marrow derived immune cells in Response to peste des Petits ruminants Virus.
While the Overall study merits publications, some technical Details have to be clarified in advance. The authors use Balb/c mice. Although this is an inbred strain they should Show what the transcriptional priofile Alteration between several non-infected animals is in order to define thresholds. This is not or not sufficiently described.
Answer for Q: 6 mice were selected for each group randomly, three groups there are 18 mice, then they were anaesthetized and killed to get the tibia. The cells were isolated from tibia and were seeded into six-well plates (3×106 cells/well). Three independent repeats.
2, Moreover it can make a difference if the mice are 6 or 8 weeks old, thus for transcriptome analyses all controllable Parameters should remain constant.
Answer for Q: Thanks for reviewers' suggestions. In our study, to be exact, the mice were eight weeks old when the mice were put to painless death.
3, The reviewer also has not found Information on the number of animals in each Group. This is also essential for the study. It is strongly recommended that an a priori animal number calculation is to be performed to justify the Group size.
Answer for Q: Thanks a lot for reviewers' suggestions. In each independent experiment, 6 mice were selected from every group randomly (Mock, LPS, PPRV), a total of 18 mice. Then they were anaesthetized and killed to get the tibia. Three independent repeats, so we used 48 mice. (See more details in line 77-79 in vetsci-646855R1)

Round 2
Reviewer 1 Report
Thanks for responding to points during review. Keep it up!
Reviewer 4 Report
my previous comments have been fully addressed, thank you very much. The study appears to be Sound now, no further requests from my Point of view.